# Terrestrial reproduction and parental care drive rapid evolution in the trade-off between offspring size and number across amphibians

**Andrew I. Furness** [1,2]*, **Chris Venditti** [3], **Isabella Capellini** [4]*

**1** Department of Biological and Marine Sciences, University of Hull, Hull, United Kingdom, **2** Energy and Environment Institute, University of Hull, Hull, United Kingdom, **3** School of Biological Sciences, University of Reading, Reading, United Kingdom, **4** School of Biological Sciences, Queen's University Belfast, Belfast, United Kingdom

* afurness001@gmail.com (AIF); I.Capellini@qub.ac.uk (IC)

**Data Availability Statement:** The authors confirm that all data underlying the findings are fully available without restriction. The dataset compiled and analysed for this manuscript has been

## Abstract

The trade-off between offspring size and number is central to life history strategies. Both the evolutionary gain of parental care or more favorable habitats for offspring development are predicted to result in fewer, larger offspring. However, despite much research, it remains unclear whether and how different forms of care and habitats drive the evolution of the trade-off. Using data for over 800 amphibian species, we demonstrate that, after controlling for allometry, amphibians with direct development and those that lay eggs in terrestrial environments have larger eggs and smaller clutches, while different care behaviors and adaptations vary in their effects on the trade-off. Specifically, among the 11 care forms we considered at the egg, tadpole and juvenile stage, egg brooding, male egg attendance, and female egg attendance increase egg size; female tadpole attendance and tadpole feeding decrease egg size, while egg brooding, tadpole feeding, male tadpole attendance, and male tadpole transport decrease clutch size. Unlike egg size that shows exceptionally high rates of phenotypic change in just 19 branches of the amphibian phylogeny, clutch size has evolved at exceptionally high rates in 135 branches, indicating episodes of strong selection; egg and tadpole environment, direct development, egg brooding, tadpole feeding, male tadpole attendance, and tadpole transport explain 80% of these events. By explicitly considering diversity in parental care and offspring habitat by stage of offspring development, this study demonstrates that more favorable conditions for offspring development promote the evolution of larger offspring in smaller broods and reveals that the diversity of parental care forms influences the trade-off in more nuanced ways than previously appreciated.

## Introduction

Life history theory [1,2] aims to explain how diverse life history strategies evolve under natural selection. Central to the theory are trade-offs that optimize resource allocation to the competing demands of growth, reproduction, and self-maintenance, under the assumption that individual resources are finite. A key life history trade-off is between the number and size of

uploaded as S1 Data. The sources for the data are available in S1 Data and list in S3 Data. The phylogeny pruned from Pyron (2014) and used for the analysis is uploaded as S2 Data.

**Funding:** We thank the University of Hull and Queen's University Belfast for supporting this project with funding to IC, and Leverhulme Trust (Research Project Grant RPG-2017-017 to CV) for funding this research. The funders had no role in study design, data collection and analysis, decision to publish, or preparation of the manuscript. Leverhulme Trust: https://www.leverhulme.ac.uk.

**Competing interests:** The authors have declared that no competing interests exist.

**Abbreviations:** ESS, effective sample size; MCMC, Markov chain Monte Carlo; PGLS, phylogenetic generalized least squares; VIF, variance inflation factor.

offspring produced in a given reproductive attempt [3,4]. In many species, larger offspring are of greater quality and enjoy higher survival [5,6]. However, producing larger offspring comes at the cost of having fewer of them [1,2], a theoretical prediction repeatedly supported in many animal and plant species and populations [5,7,8]. Surprisingly, despite much research, it is still unclear whether and how selective pressures related to environmental conditions and type of parental care drive evolutionary changes in the offspring size–number trade-off [9–20]. Answering this question is fundamental not only for advancing theory, but also because these life history traits influence the demographic trajectory of natural and introduced populations [21–25] and their ability to overcome many anthropogenic stressors [24,26–28]. For example, in several taxa, fecundity influences extinction risk [27,29], population growth rate [30], invasion success [21,22], and the ability to thrive in urban, or more generally human-modified, habitats [26,28]. Here, we test hypotheses predicting that parental care or terrestrial habitats in which offspring develop alter the offspring size–number trade-off, specifically leading to smaller clutches of larger eggs, in a sample of over 800 amphibian species. Importantly, selection is expected to increase rates of phenotypic evolution on target traits [31–36]. Therefore, we expect that egg and clutch size exhibit higher phenotypic change when under selection imposed by parental care and offspring habitat. To test these predictions, we employ cutting edge Bayesian phylogenetic comparative methods [31,37] that quantify the strength and direction of associations of parental care and offspring habitat with the trade-off and simultaneously identify heterogeneity in rates of phenotypic evolution. By testing whether higher rates of evolution in egg or clutch size are explained by parental care diversity and offspring habitat, our approach allows us to get closer to causation at a large comparative scale than what is possible with standard phylogenetic comparative methods. Finally, unlike most previous studies, we treat individual care behaviors and adaptations by each parental sex as separate drivers rather than clumping them together, since they likely entail different costs and benefits to the carer (s), are likely favored under diverse ecological conditions, and may thus influence the trade-off differently. Combined, our analyses provide a powerful and comprehensive test of the hypotheses that parental care diversity and offspring habitat are responsible for evolutionary changes in egg and clutch size.

Natural selection should favor the evolution of larger eggs when environmental conditions are favorable and offspring survival is high. Specifically, theoretical models and empirical studies find that females invest in larger offspring when predation on offspring is low, or in stable, competitive environments [18,38–41]. Terrestrial habitats may promote the evolution of larger eggs in amphibians through different mechanisms. Since eggs are eaten by many vertebrate and invertebrate species in aquatic habitats, laying eggs on land has long been viewed as an adaptation that minimizes egg predation and could in turn promote the evolution of larger eggs (fish: [42]; amphibians: [39,43–45]). Moreover, larger eggs in terrestrial habitats experience lower water loss, hence risk of desiccation, having a more favorable volume to surface ratio than smaller ones [46,47].

Parental care is also considered a possible driver for the evolution of larger offspring. Specifically, theoretical models suggest that parental care may evolve to buffer the offspring against unfavorable environmental conditions when offspring mortality is high or in favorable but ephemeral habitats [13,48–53]. Subsequently, higher offspring survival should promote additional parental investment, such as in larger eggs [13,52,53]. This is because, while larger eggs take longer to hatch and thus require prolonged parental protection, they are of higher fitness value to parents as they result in larger larvae or juveniles that suffer low mortality and reach sexual maturity early [20].

Offspring number rather than offspring size may, however, be the target of selection. Environmental conditions can drive brood size evolution [19,27] since they determine the

mortality risks on eggs and young, for example, due to predation, desiccation, thermoregulation, or oxygen availability [17,20,54,55], and the amount of resources available to females for provisioning their eggs and young posthatching/birth in caring species [17]. Discovering which trait in the trade-off is under selection is thus not trivial because reasons as to why females should invest in larger or in more offspring may differ and a change in one of the traits in the trade-off does not always lead to a proportional change in the other. For example, mammals with biparental care are more fecund, but neonatal size does not differ in species with biparental or uniparental care, suggesting that biparental care has evolved to enhance parental, rather than offspring, fitness [56]. Similarly, selection in reef fishes has acted differently on egg size and clutch size [18]. Specifically, demersal guarded eggs are larger than pelagic and scattered eggs as predicted by theoretical models of parental care evolution [13,52,53], while clutches are bigger in larger species, likely reflecting fecundity selection [19,27], but clutch size is unrelated to egg size. To identify whether selection by a proposed driver alters offspring size and/or offspring number, both elements of the trade-off, needs to be considered.

Comparative studies on the relative importance of the drivers of offspring size and number reach different conclusions. For example, some found that eggs are larger in caring than noncaring fish species [11,13,20], but egg size is unrelated to parental care in insects [9] and neonatal size is unrelated to any male care behaviors in mammals [56]. Likewise, studies in amphibians disagree on whether care, terrestrial egg development, or neither is associated with larger eggs or smaller clutches [10,12,14–16]. Many previous comparative studies have focused only on offspring size and care, or offspring number and environmental conditions, and ignored one or more of the following factors that may covary with them: the trade-off between offspring size and number as discussed above, the stage of development (egg, larvae, or juvenile) at which the offspring are terrestrial (offspring habitat) and/or are cared for, the diversity in parental care strategies, and allometry (S1 Table). Specifically, offspring size may change not because directly under selection but when selection alters offspring number. Likewise, female size may change over evolutionary time due to selection unrelated to reproduction (for example, predation risk, resource availability and interspecific resource partitioning, thermoregulation [57–61]). Since larger females typically produce larger offspring and are more fecund in many taxa [19], a change in female size may only indirectly affect offspring size and/or number even if neither is directly under selection [10,62,63]. Moreover, amphibian terrestrial eggs are larger than aquatic eggs and frequently cared for [10,16,64]. Finally, direct developing eggs (i.e., those hatching as juveniles) are larger than those hatching as tadpoles in amphibians because they require more resources to complete development [65,66], are generally terrestrial, and laid in smaller clutches [10]. Thus, we need to consider all these factors simultaneously if we are to disentangle how strongly parental care and offspring habitat affect the evolutionary trajectory of offspring size and/or number.

While parental care strategies are highly diverse, previous comparative studies have typically reduced care to a simple binary trait (care or no care). This is problematic because the costs and benefits of care likely differ between the sexes [67] and vary across care forms. For example, in mammals, 2 male care behaviors, carrying the offspring and provisioning the mother, are associated with more frequent breeding and larger litters, respectively, while 2 other paternal behaviors, grooming and huddling with the offspring, are unrelated to life history traits [56]. Finally, diverse ecological and social conditions may promote distinct care behaviors and adaptations. In amphibians, tadpole feeding is associated with larval development in small, water-filled, plant cavities [68], while care forms at the tadpole and juvenile stage are promoted by the earlier evolution of egg attendance in both sexes [69]. Hence, clumping care diversity into a simple binary trait likely obscures meaningful differences in how diverse care forms, ranging from simple egg attendance to complex morpho-physiological

adaptations like viviparity, may influence the trade-off between offspring size and number. Recent studies have attempted to incorporate diversity in amphibian parental care by ranking care forms for their presumed degree of protection or nutrition (S1 Table); however, such classifications have limited empirical foundation.

Amphibians are an ideal group in which to investigate how offspring habitat and diverse forms of parental care influence the offspring size–number trade-off. While the majority of amphibians spawn in aquatic environments and provide no care, an incredible diversity of care forms at the egg, tadpole, and/or juvenile stage of offspring development has evolved (and been lost) multiple times across the phylogeny and is found in about 25% of extant species [69]. These include attendance, transport, brooding (eggs or tadpoles develop on or inside the parental body), feeding (the mother provisions the offspring with trophic eggs or sloughed-off skin), and viviparity [69] (Table 1). With the exception of feeding and viviparity, all these care behaviors and adaptations can be found in both sexes in amphibians [69] (Table 1). Moreover, eggs, larvae, and/or juveniles can be terrestrial or aquatic and eggs may hatch as either tadpoles or juveniles (i.e., direct development). Using this diversity, we investigate the hypotheses that parental care and terrestrial offspring habitat select for larger eggs and smaller clutches. We thus test theoretical predictions [13,39,40,52,53] for expected positive associations between egg size, parental care forms, and terrestrial habitat at each stage of development (egg, tadpole, and juvenile), while controlling for allometry, clutch size, and direct development. We build similar models swapping clutch size and egg size when investigating the evolution of clutch size.

Importantly, the development of new cutting-edge phylogenetic methods and large phylogenies with hundreds of species now offer the opportunity to test the novel prediction that the rates of egg and clutch size evolution increase if they are under selection imposed by the evolution of parental care and/or terrestrial habitat. It is well documented that the speed at which

**Table 1. Description of amphibian parental care.** Brief definition of parental care forms from Furness and Capellini [69], where details of the data collection protocols can be found. Sample sizes for the variables as used in this study can be found in S2 Table. All care forms with the exception of tadpole feeding and viviparity can be carried out by either (or both) sexes in a species.

| Parental care form (caring sex) | Definition |
|---|---|
| **Egg attendance** (♀ or ♂) | A parent remains (full or part-time) with the eggs at a fixed location. |
| **Egg brooding** *(♀ or ♂)* | A parent broods the eggs on (for example, in pouches on the back, between the hindlegs) or inside their body (for example, vocal sacs, stomach, under the dorsal skin). |
| **Tadpole attendance** (♀ or ♂) | A parent remains (full or part-time) with the larvae (aquatic or terrestrial). |
| **Tadpole transport** (♀ or ♂) | Relocation of tadpoles from one habitat to another, where they become free-living. |
| **Tadpole brooding** (♀ or ♂) | Tadpoles complete most or all of their development inside or on the body of the parent and are not free-living. |
| **Tadpole feeding** *(♀)* | Female provides eggs for tadpoles to consume (Anura). |
| **Juvenile attendance** (♀ or ♂) | A parent remains (full or part-time) with juveniles at a fixed location. |
| **Juvenile transport** (♀ or ♂) | Transport of newly hatched froglets on parent's body. |
| **Juvenile feeding** (♀) | Female provisions juveniles with sloughed off skin (i.e., dermatophagy or skin-feeding in Caecilians). |
| **Viviparity** (♀) | Female gestates offspring in the oviducts and gives live birth. This category includes lecithotrophic and matrotrophic viviparity. |

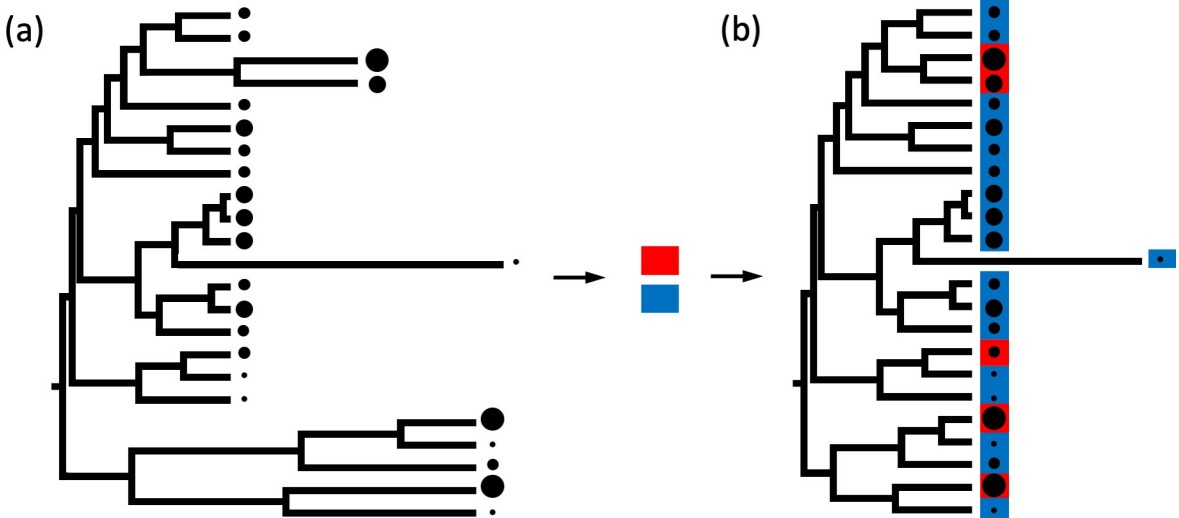

**Fig 1. Predictions linking rate heterogeneity to proposed selective drivers using phylogenetic variable rate models.** We first identify heterogeneity in rates of phenotypic evolution in a target trait (i.e., egg or clutch size) with a simple model (A) that excludes any predictor of interest. The simple model identifies branches with exceptionally high rates of evolutionary change, which are visually represented as stretched branches. As an example, here, we illustrate egg size (represented by size of the black dots). Our simple model for egg size included only body size and clutch size (these predictors are not visually represented in this figure). Next, we run a model including additional explanatory variable(s) of interest (for example, parental care forms, offspring habitat, direct development). If the additional predictor(s) select for changes in egg size, they should explain at least some of the rate heterogeneity observed in the simpler model (A). Thus, variable rate models including additional predictor(s) are expected to identify fewer branches with exceptional rates of egg size evolution (B) when compared to the simpler models without them (A). For simplicity, in this example, we visualize the expected effect of only one predictor of interest, offspring habitat (red, terrestrial; blue, aquatic), on egg size (black dots; larger dots indicate larger eggs).

traits evolve (the "rate of phenotypic evolution") may differ across a phylogeny ("rate heterogeneity"). In other words, rate heterogeneity indicates that phenotypic traits have accumulated more (or less) change along some branches of a phylogeny than expected for their length (often measured as time) and when compared to other branches. Given that phenotypic traits should evolve more rapidly when under intense selection [31–37,70–72], high rates of phenotypic evolution in some branches could be the result of stronger selection in those branches. Here, we use variable rates models [37] (Methods, Variable rates model) not only to quantify the direction and magnitude of effects of proposed drivers (parental care, offspring habitat, and direct development) on target traits (egg and clutch size) like standard phylogenetic approaches (such as phylogenetic generalized least squares (PGLS) models), but also to identify branches in the phylogeny along which egg and clutch size have accumulated exceptionally high levels of phenotypic change. Such branches are visually represented as stretched branches and indicate episodes of likely strong selection (Fig 1A). If the proposed drivers are responsible for intense selection on target traits, causing exceptional evolutionary rates, we expect that they explain at least some rate heterogeneity identified by variable rates models in which the proposed drivers are not included. Therefore, the proposed drivers should associate with target traits in the direction predicted by the hypothesis (as in PGLS), and, in addition, models including them as predictors should exhibit fewer (still unexplained) cases of exceptional rates (stretched branches) compared to those in models without them (Fig 1A versus 1B). In the context of this study, we should therefore find that, if parental care and terrestrial habitat select for larger eggs and/or smaller clutches, stretched branches indicating exceptional evolutionary rates in egg or clutch size are fewer in models with care and offspring habitat included as predictors relative to models without them (Fig 1A and 1B).

## Results

### Egg size

We use phylogenetic variable rate models and a sample of over 800 species with no missing data (S1 Data; sample sizes in S2 Table) to quantify the relative importance of individual care forms and offspring habitat on egg size, while accounting for the trade-off with clutch size, allometric effects, and direct development. Our models simultaneously identify branches in the phylogeny exhibiting exceptional rate of egg size evolution. Starting with a model including all predictors ("Full model," S3A Table), we followed a model simplification procedure, progressively eliminating each nonsignificant predictor until only significant predictors remained ("Reduced model," S3B Table) (Methods, Identifying significant predictors of egg and clutch size evolution). Consistent with theoretical predictions [40,52,53], our variable rates approach reveals that eggs are larger in species with larger body size, smaller clutches, terrestrial eggs, direct development, egg brooding, female egg attendance, and male egg attendance (Fig 2A, S3B Table). In contrast to predictions, however, eggs are smaller with 2 female care behaviors: tadpole feeding and female tadpole attendance (Fig 2A, S3B Table).

We further evaluate the relative importance of each significant predictor (S4A Table) by examining the change in the model's marginal likelihood when one predictor at a time is removed, which provides an estimate of effect sizes for each predictor (Methods, Identifying significant predictors of egg and clutch size evolution). This shows that body size, clutch size, and direct development have a greater effect on egg size than terrestrial eggs and parental care

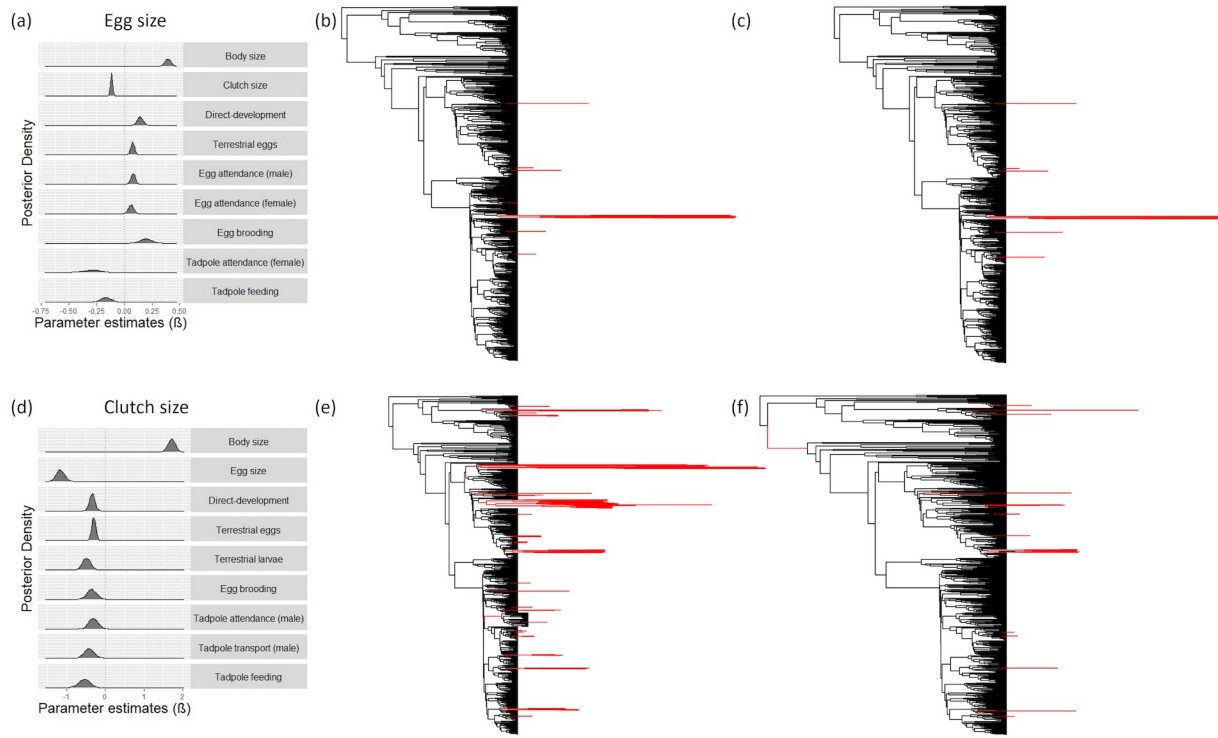

**Fig 2. Egg and clutch size evolution.** In (A) and (D), posterior distributions of the parameter estimates (β) of the significant predictors in the reduced model (S3B and S3D Table) for egg size (A) and clutch size (D), using phylogenetic variable rates models. Branches of the amphibian phylogeny that exhibit exceptional rates of evolution ($r > 1$ in ≥95% of the posterior distribution) are depicted in red for egg size (simple model, with only body size and clutch size as predictors, in (B), and reduced model in (C)) and for clutch size (simple model, with only body size and egg size as predictors, in (E), reduced model in (F)). The identity of these branches is reported in S5 Table for egg size and S6 Table for clutch size. Raw data for these analyses are available in S1 and S2 Data.

forms (S4A Table). These effect sizes correspond to a remarkable increase in egg size for an average-sized amphibian with average clutch size, ranging between approximately 15% for female egg attendance to 57% for egg brooding, and a decrease in egg size of nearly 50% with female tadpole attendance and 32% with tadpole feeding (S4A Table).

Variable rates analysis also identifies exceptional rates of egg size evolution, while accounting only for body size and clutch size, in 19 branches (Fig 2B, S5 Table) (Methods, Identifying rate shifts). These branches indicate the phylogenetic position of episodes of intense selection on egg size. The significant predictors in the reduced model (S3B Table) explain the exceptional rates of egg size evolution in only 3 of these branches (Fig 2B and 2C, S5 Table).

## Clutch size

From the full model with all predictors, model simplification procedure identifies the significant predictors. The reduced model with only significant predictors from our variable rates analysis demonstrates that clutches are bigger in larger species and smaller with larger eggs, terrestrial eggs, terrestrial larvae, direct development, egg brooding, male tadpole attendance, male tadpole transport, and tadpole feeding (Fig 2D, S3C and S3D Table). Changes in model marginal likelihood when one predictor at a time is removed indicate that after body size and egg size, terrestrial eggs, terrestrial larvae, and direct development have a greater effect on clutch size than care forms (S4B Table). These effect sizes correspond to a reduction in clutch size, ranging from about 50% with terrestrial eggs and male tadpole attendance up to 71% with tadpole feeding for an average-sized amphibian producing eggs of average size (S4B Table). While accounting for body size and egg size, we identify exceptional rates of clutch size evolution in 135 branches, 108 of which (80%) are explained by care forms, direct development, and terrestrial eggs and larvae (Fig 2E and 2F, S6 Table). This suggests that offspring habitat, direct development, and care forms are responsible for intense selection on clutch size in these branches.

## Discussion

The trade-off between offspring size and number is central to life history theory and has important implications in both basic and applied questions; however, which selective pressures influence its evolution, is debated. Here, we have investigated 2 hypotheses proposing that parental care and/or more favorable terrestrial habitats for offspring development select for larger eggs [13,39,40,52], while accounting for allometric effects, direct development, and the trade-off with clutch size. We have also asked whether the proposed drivers have acted on clutch size rather than egg size. Our results show that amphibians with direct development and those with terrestrial offspring have both larger eggs and smaller clutches (Fig 3). Considering individual parental care forms separately has allowed us to unravel the complex and contrasting influence that they exert on the trade-off, with egg brooding, male and female egg attendance increasing egg size, female tadpole attendance and feeding decreasing egg size, and egg brooding, tadpole feeding, male tadpole attendance, and male tadpole transport reducing clutch size (Fig 3). Importantly, by simultaneously considering variation in rates of phenotypic evolution across the phylogeny, our variable rates analyses demonstrate that the significant predictors of egg and clutch size evolution can explain much of the rapid phenotypic change in these life history traits, indicating that they have imposed intense selection on the offspring size–number trade-off.

We find broad support for theoretical models that both parental care and terrestrial offspring habitat promote the evolution of larger offspring, but the role of parental care is more complex than previously appreciated and depends on the type of care, the stage at which care

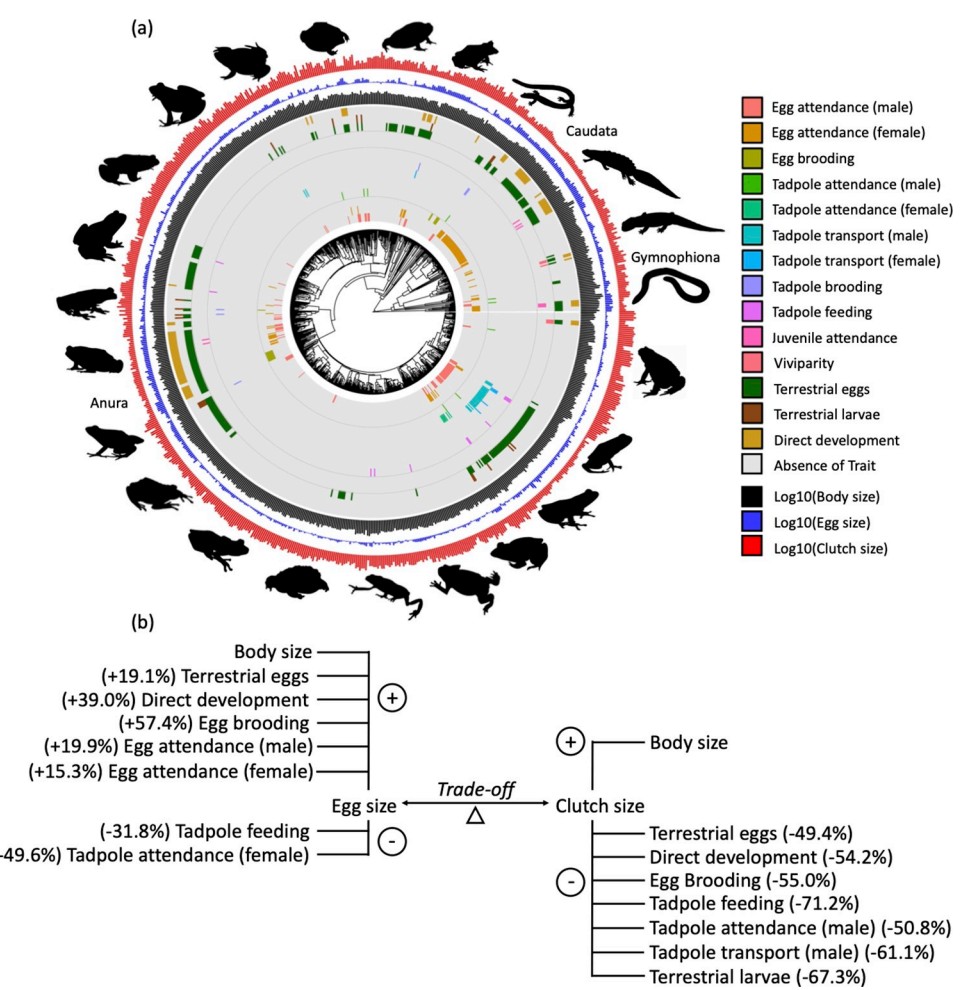

**Fig 3. Summary of results on the evolution of egg size–clutch size trade-off.** (A) Distribution of parental care forms, offspring habitat, direct development, and life history traits in amphibians (*n* = 805 species; raw data in S1 and S2 Data). All variables, except clutch size, egg size, and body size, are binary. (B) Summary of the significant associations for the trade-off between egg size and clutch size, combining their respective reduced models (S3B and S3D Table). Variables associated with increases in egg or clutch size are above the trade-off and indicated with a plus; variables associated with decreases in egg or clutch size are below the trade-off and indicated by a minus. For each variable, we report in brackets the percentage of change in egg or clutch size computed for an average-sized amphibian with average clutch size or egg size, respectively (S4A Table for egg size and S4B Table for clutch size).

is provided and the sex of the caring parent. Specifically, while accounting for allometry and the trade-off with clutch size, our study demonstrates that eggs are larger by about 20% if terrestrial, as predicted by theoretical models suggesting that favorable environmental conditions for offspring development select for larger offspring [40]. In support of the theoretical prediction that parental care drives an increase in egg size [52], we simultaneously find that eggs are 15% and 20% larger if attended by females and males, respectively, and nearly 60% bigger if brooded by parents. However, in contrast to this prediction [52], eggs are smaller by 32% with tadpole feeding and by nearly 50% with female tadpole attendance. Although these results may seem unexpected, we note that Nussbaum and Schultz's theoretical model [40] predicts that, at any given level of parental care, egg size may decrease if environmental conditions for juvenile survival improve. We propose that this may be the case for female tadpole attendance, which occurs in ponds and terrestrial protected habitats, such as burrows, where the tadpoles can be

defended against predators [39,73]. We suggest that tadpole feeding females may not need to produce large eggs because they continue to provision their young throughout larval development, analogous to matrotrophic viviparous fish (for example, those with maternal provisioning of offspring via a placenta). Oviparous and viviparous species without matrotrophy typically supply their eggs fully before fertilization, after which they provide no further nutrition [74,75]. Conversely, females of matrotrophic fish start off with small eggs, which they continue to provision throughout development [74,75]. Overall, accounting for diversity in parental care has allowed us to unravel that different care behaviors and adaptations may drive the evolution of egg size in different directions and at different magnitude. We anticipate that similar results will be found in taxa with high diversity of care forms, such as other vertebrate classes, insects, and crustaceans.

Theoretical models and empirical studies on the role of care and offspring habitat have primarily focused on the evolution of offspring size alone [14,15,40,52,53]. However, selection may act on offspring number instead and only indirectly alter offspring size. While accounting for offspring number in statistical models offers a much stronger test of hypotheses on proposed drivers of offspring size evolution, investigating whether such drivers also affect offspring number (while accounting for offspring size) provides a comprehensive answer. We thus also ask whether the proposed drivers of egg size evolution directly affect clutch size. After accounting for allometry and the trade-off with egg size, our variable rates analysis indicates that terrestrial habitat at the egg and larval stage, direct development, egg brooding, tadpole feeding, male tadpole attendance, and male tadpole transport are associated with a substantial reduction in clutch size ranging between 50% and 70%. These results are consistent with numerous physiological and physical mechanisms known to constrain clutch size. For example, because oxygen diffusion is compromised within the jelly of terrestrial eggs, smaller clutches ensure sufficient oxygenation by reducing competition [46]. Oxygen limitation is likely to also be particularly acute for direct developing eggs, typically laid on land, given their extended period of development and large size. For tadpole feeding, we suggest that mothers cannot support large clutches because they often provide energetically expensive nutrition over a long period of offspring development. Consistent with this idea, female strawberry poison frogs (*Oophaga pumilio*) lay fewer eggs when simultaneously provisioning older tadpoles, while tadpoles in larger clutches receive smaller meals and suffer higher mortality [76]. Instead, physical space may constrain clutch size in brooding species, in species with male tadpole transport, and those with terrestrial tadpoles. Specifically, the size of the body cavity or surface area of the back is likely to limit the number of eggs parents can care for in egg brooding frogs or the number of tadpoles that males can transport [73]. Likewise, terrestrial tadpoles with no caring parents frequently develop in foam nests, burrows, or within cup-shaped nests and typically do not feed [39,64]. These confined spaces are likely to provide shelter to only a few tadpoles, while limitation to oxygen diffusion might further constrain the number of developing larvae as it does for eggs. While constraints on offspring number beyond amphibians have been previously discussed mostly in relation to viviparity [11,77], our results suggest that clutches are likely to be reduced in many other species in which the eggs or young are physically associated with the parental body or are placed in microenvironments or nests where physiological or physical conditions impose an upper limit to the number of offspring they can accommodate.

Bringing findings for egg size and clutch size together, this study reveals how proposed drivers affect both or only one of the 2 elements of this trade-off (Fig 3B). Specifically, terrestrial eggs, egg brooding, and direct development act simultaneously on both egg and clutch size, i.e., directly increase egg size and decrease clutch size, beyond the indirect effect that they already have on the other element of the trade-off. Instead, male egg attendance, female egg

attendance, and female tadpole attendance only associate with egg size; tadpole terrestriality, male tadpole attendance, and male tadpole transport only associate with smaller clutches; while tadpole feeding is associated with both smaller eggs and smaller clutches. By accounting for diversity in care forms and offspring habitat by stage of development, we demonstrate that terrestrial habitat and direct development consistently lead to larger eggs and smaller clutches, while different care forms can have contrasting effects on the offspring size–number trade-off. Thus, it is not surprising that studies clumping or arbitrarily ranking care forms across developmental stages find that terrestrial habitat, but not parental care, is associated with egg size or number. Many theoretical models consider parental care as a uniform species characteristic that varies in duration, intensity (for example, how much food to provision), or, in the context of sexual conflict, by caring sex. However, our study demonstrates that the effect of parental care on egg size and clutch size is far more complex and differs depending on the type of care, the stage at which care is given and the sex of the carer. We thus need a new theoretical framework that explicitly considers such diversity of care and provides quantitative predictions on how different care behaviors and adaptations should impact the evolutionary trajectory of egg and clutch size and, more broadly, life history strategies.

Importantly, our variable rates models explicitly consider heterogeneity in rates of phenotypic evolution and simultaneously identify where in the phylogeny egg size and clutch size have accumulated more phenotypic change than expected, indicative of intense selection. Episodes of exceptional rates of egg size evolution are few (19 branches), and the significant ecological and parental care predictors account for only 3 of these exceptional rates. In contrast, exceptional rates of evolution in clutch size were frequent (135 branches), and 80% of these were explained by ecological and parental care predictors. Thus, our approach reveals that many more branches across the phylogeny exhibit higher rates of phenotypic evolution for clutch size than egg size, most of which is explained by offspring habitat, direct development, and care forms. This likely reflects the potential physiological constraints on the size of anamniotic eggs (for example, due to oxygenation [46,47]) and the higher interspecific variance in clutch size (ranging from 1 to tens of thousands) than egg size. Therefore, our study reveals that there is greater opportunity for selection on clutch size than egg size in amphibians. Based on our findings, we expect that selection has acted more strongly on offspring number than on offspring size in lineages with high diversity of parental care forms and adaptations, high diversity of habitats in which the eggs develop, and large variance in clutch size, like fish and insects. Conversely, egg size may be under stronger selection than clutch size in lineages like birds that exhibit lower diversity in care forms compared to amphibians and are limited in the number of offspring they can produce due space limitations within nests. We suggest that future comparative studies testing hypotheses on the evolutionary drivers of this key life history trade-off consider both offspring number and size and explicitly incorporate diversity in parental care, while theoretical models should evaluate under which conditions the greater response of clutch size to selection affects the evolutionary trajectory of offspring size.

To conclude, this study demonstrates that evolutionary changes in offspring habitat, parental care, and direct development have led to rapid adaptive evolution in egg and clutch size. While terrestrial offspring habitat influences the offspring size–number trade-off as predicted by theoretical models [13,39,40,52], considering the full diversity in care forms by stage of offspring development and sex of caring parent has revealed that different care behaviors and adaptations have contrasting effects on the trade-off. Importantly, incorporating variation in rates of egg and clutch size evolution in our theoretical framework has allowed us to test predictions not only on the direction and magnitude of effects of proposed drivers, but also on how proposed drivers lead to rapid change in the trade-off. Our approach thus reveals that episodes of rapid evolution in egg and clutch size are explained by offspring habitat, direct

development, and some care forms, as expected if these traits select for rapid adaptive changes in egg and clutch size to new conditions. More broadly, we expect that other comparative studies incorporating rate heterogeneity in their theoretical and analytical framework will further reveal how behavioral traits and ecological conditions explain rapid phenotypic change, and thus identify episodes of intense selection, at a large comparative scale.

## Methods

### Data collection

Parental care and direct development. Parental care data (attendance, transport, brooding, feeding, viviparity) at the egg, tadpole, and juvenile stage (Table 1) were taken from Furness and Capellini [69] where detailed descriptions of the data collection protocols can be found. All parental care variables are binary (S1 Data). In our analyses, we included all forms of care that were represented by more than 5 species exhibiting the trait of interest (S2 Table), hence we discarded juvenile transport and juvenile feeding. Likewise, we considered the sex of the caring parent in each care form only if the number of species exhibiting a trait remained greater than 5 (S2 Table).

Direct development referred to eggs hatching directly as juveniles (as opposed to larvae) or offspring being born as juveniles in viviparous species. We thus class all species with a larval stage as lacking direct development and those without a larval stage as having direct development, regardless of whether they were cared for or not, and, if cared for, irrespective of the form of care received and the sex of the caring parent. Thus, direct development was a binary variable (sample sizes by category in S2 Table; raw data in S1 Data).

Offspring habitat. Data on the environment where eggs and tadpoles are found were extracted from 458 primary and secondary sources and cross-checked (reference list in S1 and S3 Data). We discarded species lacking information on these variables, or for which information was contradictory between sources, and this contradiction could not be resolved. Thus, we did not infer the condition for species with ambiguous information from data of closely related species.

Unlike most previous studies, we classified the habitat where eggs and tadpoles are found (i.e., aquatic or terrestrial) separately and based on microhabitats, because the risk of desiccation may differ substantially by stage of offspring development (i.e., where only the eggs or tadpoles are aquatic) and vary by microhabitat. Therefore, we scored the habitat in which eggs are laid as a binary trait, i.e., as aquatic or terrestrial, based upon oviposition location (sample sizes in S2 Table). Eggs were scored as aquatic if they developed in water irrespective of the location or size of the water body (i.e., streams, small or large ponds, small pools), and eggs in foam nests on the water surface or in excavated basins partially filled with water or directly adjacent to water were scored as aquatic. Conversely, eggs that developed on the ground away from water (i.e., in leaf litter, in burrows or nests or soil cavities, under stones or logs, in rock crevices, and similar) or arboreally (i.e., attached to leaves or vegetation) were scored as terrestrial. Likewise, eggs located in foam nests in terrestrial subterranean or excavated chambers far removed from water were scored as terrestrial. Eggs laid in phytotelmata (plant cavities filled with water) were scored as terrestrial if the eggs were explicitly described as being placed terrestrially in phytotelmata (i.e., above the waterline on bamboo internodes, on bromeliad leaves, side of tree hole, or similar); in all other cases, such eggs were classified as aquatic. Egg brooding species (eggs developing on or inside the parental body) with terrestrial adults were scored as having terrestrial eggs, whereas brooding species of the family Pipidae in which adults live in water were classed as having aquatic eggs. Likewise, viviparous species with terrestrial adults were scored as exhibiting terrestrial eggs, and viviparous Caecilian species in the family Typhlonectidae in which adults live in water were classified as having aquatic eggs.

We classify the habitat in which tadpoles grow as a binary trait (sample sizes in S2 Table). Specifically, we considered larvae as terrestrial when they develop exclusively on land away from water in a terrestrial nest, burrow, on the ground or on rocks, or on or inside the parent's body and the adult is terrestrial. All other species had larvae developing in water, including those whose tadpoles develop in phytotelmata, or direct development and are classified as not having terrestrial tadpoles. Data on egg and larval habitat can be found in S1 Data.

Life history data. We found data on life history traits for 805 species with data on parental care, direct development, and offspring developmental habitat (S1 Data). Egg size (mm), clutch size, and adult body length (mm) were taken from Allen and colleagues [21], Oliveira and colleagues [78] (AmphiBIO), and additional primary sources (reference list in S1 and S3 Data). Egg size and clutch size were reported as a range (minimum–maximum) in Oliveira and colleagues [78] but as means in Allen and colleagues [21]. Therefore, we calculated the midpoint of the minimum and maximum values for egg and clutch size in AmphiBIO and combined with Allen and colleagues' data [21] (see below). We checked all egg size values of viviparous species in both AmphiBIO and Allen and colleagues' data sets. AmphiBIO sometimes recorded offspring body size at birth in viviparous species in the same column as egg diameter. These values were discarded because they are not comparable, given that the measured length of an offspring at birth (i.e., uncoiled) is necessarily larger than egg diameter and it is not taken at the same developmental point. We also discarded all egg size values for species with matrotrophic viviparity. In matrotrophic species, egg size is initially small and offspring increase in size over the course of gestation [79]; we could not verify when in development egg/offspring size was measured and therefore whether this was taken at a comparable stage to that of the oviparous taxa. Thus, the only values retained for egg size in viviparous taxa were for species that exhibited lecithotrophic viviparity or ovoviviparity and in which the value could be confirmed in a primary reference.

We visually identified outliers within each data set using trait by trait plots. We corrected any error (i.e., mis-entry from original source) and searched the literature for additional sources when we could not locate or determine the primary source; we then corrected the value if necessary. Thus, all outliers within each data set were either determined to be errors and corrected or verified as correct and left unchanged. Next, we plotted comparable life history trait values from the 2 data sets against each other and identified highly discrepant values. We checked the accuracy of the discrepant values by consulting primary and secondary sources. If the value from 1 data set was determined to be in error or likely to be in error, it was deleted and the value from the other data set was retained. For the few cases in which the life history trait value could not be verified, we took the mean value from the 2 data sets. We built our data set by taking life history values from Allen and colleagues [21] if available, and, if not, from AmphiBIO. Note, however, that the data on body size were not combined between the 2 data sets because Allen and colleagues [21] reported the mean snout vent length for all 3 Amphibian orders, while AmphiBIO reported maximum values of snout vent length for Anura, but total length for Caudata and Gymnophiona. Therefore, we used only data from AmphiBIO for body size. Finally, we added new data for species with missing values for egg size, clutch size, and body size from additional primary and secondary sources (reference list in S1 and S3 Data). Life history data were $\log_{10}$-transformed for statistical analysis.

Phylogeny. We used the phylogeny of Pyron [80], which does not have any polytomies, in all analyses as this is the most comprehensive time-calibrated tree for amphibians that was built solely with molecular data without imputation of missing taxa based on taxonomy (i.e., without molecular data). The phylogeny pruned for the species in our study is available as S2 Data.

## Correlated evolution of offspring size–number trade-off with parental care and offspring habitat

Multicollinearity between predictors. We assess the magnitude of multicollinearity between all predictors in the full model for egg and clutch size using variance inflation factors (VIFs) [81], computed on a non-phylogenetic regression using the R package *car*. This is a conservative approach because the inclusion of phylogeny typically weakens the strength of covariation between variables; therefore, VIFs are likely to be higher in nonphylogenetic than in phylogenetic models. VIF scores greater than 5 indicate likely problematic collinearity, and greater than 10 very problematic collinearity [81]. We found no evidence of problematic multicollinearity between predictors in our models of egg and clutch size evolution since all VIF values were less than 2.5 (S7 Table).

Variable rates model. We use phylogenetic variable rates models [31,37] in *BayesTraits* V.3 [82] to test for associations between life history traits (egg or clutch size), type of parental care, and offspring developmental habitat (terrestrial or aquatic eggs and larvae), while accounting for allometry, the trade-off between offspring size and number, and direct development (S2 Table). Variable rates model is an extension of PGLS models [83]. However, unlike standard PGLS, variable rates model can simultaneously account for deviation from the assumption of the underlying Brownian motion model, that the rate of evolution is constant throughout the phylogeny, i.e., there is heterogeneity in the rate of phenotypic evolution [37]. Specifically, this approach identifies branches in the phylogeny that have accumulated more or less phenotypic evolution than expected for their length (i.e., here, time) and relative to the rest of the phylogeny (i.e., the "background" rate). The model then stretches and compresses such branches in direct proportion to the observed higher or lower amount of phenotypic evolution. This makes these branches conform to the assumption of Brownian motion and thus allows a more accurate estimate of model parameters. Crucially, identifying which branches exhibit an exceptional rate of phenotypic evolution can help us identify the selective pressures responsible for these bursts of rapid adaptive change [31,37] (see Identifying rate shifts).

Variable rates models estimate the rate of evolution in the phylogenetically structured residual error of a linear model along the branches of the phylogeny. The model divides the Brownian variance of a continuous trait ($\sigma^2$) into 2 components, which are estimated simultaneously: a global background rate of evolution ($\sigma_b^2$) and rate scalars $r$ defining branch specific rate shifts relative to the background rate [37]. Together, the global background rate and branch specific $r$ optimize the variance for each branch and identify the branches that have experienced a higher ($r > 1$) or lower ($0 \leq r < 1$) rate of phenotypic evolution than the global background rate. This can be visually represented with a scaled phylogeny where each branch length has been multiplied by its specific scalar $r$, such that longer branches depict faster evolutionary rates (i.e., more phenotypic evolution than expected) and shorter branches slower evolutionary rates (i.e., less phenotypic evolution) than the global background rate. The model is implemented in a Bayesian Markov chain Monte Carlo (MCMC) framework with reversible jump and returns a posterior distribution of partial regression coefficients for all predictors in the model, branch specific scalars $r$, global background rate $\sigma_b^2$, and $\lambda$ (the phylogenetic signal ranging between 0 and 1 [83]). Following previous studies [37,71], we use a gamma prior ($\alpha = 1.1$, $\beta$ rescaled to give a median of 1) for the scalar parameter as this ensures that both rate increases and decreases are equally proposed. Note that we provide no a priori information to the model about how many branches should be rescaled, which branches should exhibit rate shifts, and by how much. Instead, reversible jump allows the algorithm to propose and estimate any number of scalars $r$ between 0 and the total number of nodes in the phylogeny (including the tips), anywhere in the phylogeny, as appropriate to the data [37].

All MCMC chains were run for a total of 200.5 million iterations with the first 500,000 discarded as burnin and sampling every 100,000 iterations thereafter. We used uniform priors ranging between −100 and 100 for all estimated regression coefficients and a uniform prior ranging between 0 and 1 for λ. We ensured that the effective sample size (ESS) for all estimated parameters was greater than 1,000, calculated with the R package LaplacesDemon, and visually inspected the trace plots to confirm that the chains converged and had good mixing.

Identifying significant predictors of egg and clutch size evolution. We entered egg or clutch size as the response variable in a variable rates model, and body size, clutch size, or egg size, respectively, type of parental care (female egg attendance, male egg attendance, egg brooding, female tadpole attendance, male tadpole attendance, female tadpole transport, male tadpole transport, tadpole brooding, tadpole feeding, juvenile attendance, and viviparity), direct development, and offspring developmental habitat (terrestrial eggs, terrestrial larvae) as predictors. Our data set consisted of 805 species with no missing data for any of the above 16 variables (S1 Data; sample sizes for each binary predictor are given in S2 Table). We first fit a "full model" including all predictors (S3A and S3C Table). Next, we followed a model simplification procedure starting from full models with all predictors and progressively eliminating the least significant predictor and rerunning the analysis until only significant predictors remained in the simplest statistically justifiable model ("Reduced models" in S3B and S3D Table). The significance of each predictor was evaluated as the proportion of the posterior distribution of its *beta* value that crossed zero ($P_x$), with influential predictors having $P_x < 0.05$ [31,37]. We report the mean, median, 95% credible intervals, $P_x$, and ESS of all predictors in the full and reduced models for egg and clutch size evolution in S3 Table.

We quantified the effect size of each significant predictor in the reduced models through the change in model marginal likelihood when one significant predictor was individually removed from the reduced model. Thus, we compared the median likelihood values of the reduced model against that of a reduced model missing one predictor at a time (S4 Table). For each significant binary predictor in the reduced models (S3B and S3D Table), we also estimated the percentage change in egg or clutch size when a trait of interest was present versus when it was absent. This was computed based on the parameter estimates of the reduced models and in a species of average body size with average egg or clutch size, while holding all other binary predictors as absent (S4 Table).

Identifying rate shifts. Using variable rates model, we could simultaneously identify the significant predictors of egg and clutch size and branches exhibiting significant deviations (i.e., rate shifts) in the rate of phenotypic evolution relative to the background rate. We defined branches as showing exceptional rate shifts if their estimated scalar *r* was greater than 1 (positive shifts, i.e., higher rates of evolution) or less than 1 (negative shifts, i.e., lower rates of evolution) in 95% of the posterior distribution [31]. Branches showing exceptional rates of evolution reveal that selection has acted more strongly along those branches. Therefore, to investigate whether the significant predictors in the reduced models were responsible for any shifts in the evolutionary rates of egg or clutch size, we compared the number of branches exhibiting rate shifts in the reduced models (S3B and S3D Table) with that of simpler models that only included body size and the trade-off between egg and clutch size as predictors (Fig 1A versus 1B). Branches exhibiting rate shifts in these simpler models showed that there was significant unexplained variance in egg or clutch size rates of evolution. We expected that, if parental care forms, direct development, and/or offspring habitat explained the rate heterogeneity identified in the simpler models, the number of stretched (or compressed) branches in the reduced model should be lower than the number in their respective simpler models (i.e., comparisons between Fig 1Aa and 1B). This evidence would strongly suggest that parental care, direct development, and offspring developmental habitats have selected for rapid

adaptive changes in egg or clutch size in the stretched (or compressed) branches of the simpler models that no longer exhibit exceptional rate of evolution in the reduced models. We report the list of branches exhibiting rate shifts in the simple and reduced models for egg size and clutch size in S5 and S6 Tables, respectively.

## Supporting information

**S1 Table. Summary of previous phylogenetic comparative studies on the correlated evolution between parental care, offspring developmental environment, and/or offspring size–number trade-off in amphibians.** Here, we only report results of variables of interest to the aims of our study.
(DOCX)

**S2 Table. Sample sizes for categorical, binary, independent variables considered as predictors of egg and clutch size in this study.** The total sample size of our data set is 805 species, with complete parental care, offspring habitat, direct development, and life history data (body size, egg size, and clutch size). Here, we report the sample sizes for these 805 species for all predictors considered in this study, those in which more than 5 species exhibited the trait of interest.
(DOCX)

**S3 Table. Variable rates models for egg and clutch size.** Results for the analysis of egg size in (A) full model, including all predictor variables, and (B) the simplest statistically justifiable model (reduced model) with only significant predictors after model simplification (see Methods, Identifying significant predictors of egg and clutch size evolution). Results of analysis for clutch size in (C) full model and (D) reduced model. The columns report the ESS, the mean and median of the posterior distributions, the 95% HPD interval, and the proportion of the posterior distribution crossing zero ($P_x$) for each predictor variable in the model. We also report model $R^2$, phylogenetic signal as estimated by λ, and model marginal likelihood. ESS, effective sample size; HPD, highest posterior density.
(DOCX)

**S4 Table. Effect sizes of the significant variables in the reduced model for egg size (A) and clutch size (B) and corresponding percentage change in egg and clutch size for an average-sized amphibian.** Effect sizes are quantified as the median reduction in marginal likelihood when a given predictor is individually removed and the model rerun (see Methods, Identifying significant predictors of egg and clutch size evolution). The percentage change is computed as the difference between the estimated egg (A) or clutch size (B) when each predictor in turn is present compared to when it is absent, for an average-sized amphibian with average egg (A) or clutch size (B), holding all other predictors of the reduced models as absent. The direction of the percentage change reflects an increase (+) or decrease (−) in egg (A) or clutch size (B).
(DOCX)

**S5 Table. Rate shifts in amphibian egg size evolution from variable rates model.** We report the 19 branches that show exceptional rates of evolution in egg size, relative to the background rate, from the simple model including only body size and clutch size (left column) and the reduced model including the significant predictors (right column; the statistics for the reduced model is reported in full in S3B Table) (see Methods, Identifying rate shifts). Each branch is identified by its descendants. For each branch, we also report the median of the scalar *r* (see Methods, Identifying rate shifts). These branches correspond to those highlighted in red in Fig 2B and 2C. The blank cells listed under the reduced model are those that exhibit rate shifts in

the simple model but not the reduced model, i.e., branches for which rapid egg size evolution can be attributed to the addition of parental care and reproductive ecology variables. The addition of terrestrial eggs, direct development, and parental care variables in the reduced model explains rapid egg size evolution in only 3 branches, that of *Bufo valliceps*, *Eleutherodactylus alticola*, and the clade consisting of *Centrolene geckoideum*, *Centrolene savagei*, *Cochranella euknemos*, *Cochranella granulosa*, *Espadarana prosoblepon*, *Teratohyla midas*, and *Teratohyla spinosa*.
(DOCX)

**S6 Table. Rate shifts in amphibian clutch size evolution from variable rates model.** We report the 135 branches that show exceptional rates of evolution in clutch size, relative to the background rate, from the simple model including only body size and egg size (left column) and the reduced model also including the significant care and reproductive ecology predictors (right column; the statistics for the reduced model is reported in full in S3D Table). Each branch is identified by its descendants. For each branch, we also report the median of the scalar *r* (see Methods, Identifying rate shifts). These branches correspond to those highlighted in red in Fig 2E and 2F. The blank cells listed under the reduced model are those that exhibit rate shifts in the simple model but not the reduced model, i.e., branches for which rapid clutch size evolution can be attributed to the addition of parental care and reproductive ecology variables. The addition of direct development, offspring habitat, and parental care variables explains rapid clutch size evolution in 108 branches.
(DOCX)

**S7 Table. VIFs for the full models for egg size (A) and clutch size (B).** VIF, variance inflation factor.
(DOCX)

**S1 Data. Raw data set for the 805 species used in this study.**
(XLSX)

**S2 Data. Pruned phylogeny from Pyron [80] used in this study.**
(TREES)

**S3 Data. Supplementary data references for the raw data reported in S1 Data.**
(DOCX)

## Acknowledgments

We thank Joanna Baker for sharing her R code; VIPER High Performance Computing facility and its support team at the University of Hull; James Gilbert for comments on early results of this study; and David Reznick, Martha Crump, and Jesse Delia for comments on an earlier draft of this manuscript.

## Author Contributions

**Conceptualization:** Andrew I. Furness, Chris Venditti, Isabella Capellini.

**Data curation:** Andrew I. Furness.

**Formal analysis:** Andrew I. Furness, Chris Venditti, Isabella Capellini.

**Funding acquisition:** Isabella Capellini.

**Investigation:** Andrew I. Furness, Chris Venditti, Isabella Capellini.

**Methodology:** Chris Venditti, Isabella Capellini.

**Project administration:** Andrew I. Furness, Isabella Capellini.

**Resources:** Andrew I. Furness, Isabella Capellini.

**Software:** Chris Venditti.

**Supervision:** Isabella Capellini.

**Visualization:** Andrew I. Furness, Chris Venditti, Isabella Capellini.

**Writing – original draft:** Andrew I. Furness, Isabella Capellini.

**Writing – review & editing:** Andrew I. Furness, Chris Venditti, Isabella Capellini.

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
