## [Editor Report · Decision Letter 0]

29 Jun 2021

Dear Dr Capellini, 

Thank you for submitting your manuscript entitled "Terrestrial reproduction and parental care drive rapid adaptation in the offspring size-number tradeoff across amphibians" for consideration as a Research Article by PLOS Biology.

Your manuscript has now been evaluated by the PLOS Biology editorial staff, as well as by an academic editor with relevant expertise, and I'm writing to let you know that we would like to send your submission out for external peer review.

Kind regards,

Roli Roberts

Roland Roberts

Senior Editor

PLOS Biology

rroberts@plos.org

---

## [Decision Letter · Decision Letter 1]

10 Aug 2021

Dear Dr Capellini,

Thank you for submitting your manuscript "Terrestrial reproduction and parental care drive rapid adaptation in the offspring size-number tradeoff across amphibians" for consideration as a Research Article at PLOS Biology. Your manuscript has been evaluated by the PLOS Biology editors, an Academic Editor with relevant expertise, and by three independent reviewers.

IMPORTANT: You'll see that all of the reviewers appreciate the study, but that they disagree about whether the advance is substantial enough to warrant publication in PLOS Biology (though reviewer #3 concedes that this is a subjective call). We put this divergence of opinion to the Academic Editor, and their advice was that the advance was indeed sufficient, and that we should give you an opportunity to revise and resubmit, addressing the specific concerns raised by reviewers #1 and #2. To help guide you in your revisions, I've pasted the Academic Editor's comments into the foot of this letter.

In light of the reviews (below), we will not be able to accept the current version of the manuscript, but we would welcome re-submission of a much-revised version that takes into account the reviewers' comments. We cannot make any decision about publication until we have seen the revised manuscript and your response to the reviewers' comments. Your revised manuscript is also likely to be sent for further evaluation by the reviewers.

We expect to receive your revised manuscript within 3 months. 

**IMPORTANT - SUBMITTING YOUR REVISION**

*Re-submission Checklist*

*Published Peer Review*

*PLOS Data Policy*

*Blot and Gel Data Policy*

Sincerely,

Roli Roberts

Roland Roberts

Senior Editor

PLOS Biology

rroberts@plos.org

REVIEWERS' COMMENTS:

Reviewer #1:

This work examines macroevolutionary patterns in egg size, egg number and forms of parental care in amphibians. This analysis seeks to extend a series of comparative analyses already specifically on this group. 

The authors argue that "Surprisingly, despite much research, it is still debated which selective pressures drive evolutionary changes in the offspring size-number tradeoff" and then cite an older theory paper and studies mostly focused on amphibians. The problem here is that, with a very strong focus on the amphibian literature, the authors seem to overestimated the novelty of their own work and underestimated the state of knowledge in the field more generally. For example, Lim et al. 2014 is only mentioned in passing but is a more taxonomically comprehensive exploration of ideas explored here (and controls for body size). Similarly, the idea and observation of systematic correlations among offspring size, developmental mode and fecundity goes back to at least Thorson (1936,1950), who showed strong covariance among all three for a range of marine invertebrate taxa - this work is not mentioned. This is just one example, the fact is that offspring size-number covariances have been explored repeatedly across a broad range of taxa. Plants similarly are neglected though seed size number trade-offs have been studied extensively. All in all, this work is a little tunnel visioned, focused on one group, which means the work is not placed in the appropriate context. 

The authors then explore 16 different life history predictors and how they affect (co)variance in offspring size and fecundity. My concern here is with such an extensive list of predictors, all categorical, it's unclear how much covariance occurs among predictors and yet many of these predictors are highly correlated. Covariance among predictors leads to less reliable estimates of their effects, this is the problem with a laundry list of categorical predictors. What's more, including 16 predictors in the absence of formal explorations of their covariances means that interpreting their effects is problematic. Overall, with such a long list of predictors, I am sorry to say I have insufficient confidence in the analyses. I suggest the authors consolidate their predictors. Doing so would also reduce the impression that this is simply a pattern hunting expedition - many of justifications for the inclusions of the predictors are weak and insufficiently motivated by formal theory. 

Overall, I think these are useful data for a well studied group but I don't think there's much that's new here and it doesn't really change our understanding of the topic beyond what has already been done. I think it's essential that the authors consolidate their predictor list according to the covariances among them and re-analyse their data accordingly 

Reviewer #2:

General comments

This is a novel and important study that takes an innovative approach to testing whether parental care and offspring habitat characteristics drive the evolution of the trade-off between egg size and clutch size, and therefore life-histories in general, in amphibians.

Using an impressively large comparative dataset they test, and largely find support for, predictions from theory that parental care is associated with the evolution of increased egg size (and survival) and smaller clutch sizes are associated with terrestrial offspring environments. 

The specific novelty here revolves around the use of Bayesian variable rates models that allows the Brownian motion approach (i.e. assuming a constant rate of evolution across the phylogeny) of standard Phylogenetic Generalized Least Squares models to be adapted to account for variation in the rate of phenotypic evolution. This allows the authors to quantify the direction and magnitude of the proposed drivers of the evolution in the focal traits, egg size and clutch size, i.e. parental care and offspring developmental environment, to identify particular branches of the phylogeny where there have been unusually high levels of phenotypic change. The reasonable assumption here is that this rate heterogeneity is likely to reflect variation in the strength of selection as a result of adaptive evolution driven by parental care and offspring environment. Hence, in contrast to standard comparative approaches this allows causation to be ascribed.

Furthermore, the approach also allows the trade-off between egg size and clutch size to be explicitly considered, which has not been done before. In addition, the impressively detailed dataset on over 800 species has lots of finer grained detail on specific forms of parental care. As a result the authors have uncovered some key insights, not least that clutch size shows more phenotypic evolution than egg size, most likely due to the greater degrees of freedom inherent in the number as opposed to the size of eggs, and that terrestrial habitats and direct development have consistently lead to the evolution of smaller clutch sizes and larger eggs but the effect of parental on these life-history traits depends on the particular form of care being considered.

In sum this is a very impressive piece of work that provides a substantial advance in terms of approach and insights. As such I do not have any major criticisms. However, I do think the framing of the work could be better in particular. 

Much is made of the safe harbour hypothesis, for example, which posits that parental care evolves to enhance egg survival in harsh environments. The authors suggest that the results largely provide support for this hypothesis based on the positive relationships between several different forms of parental care and egg size. However, there is also a positive effect of terrestrial offspring environment on egg size too. Given that terrestrial environments are deemed to be less harsh than aquatic environments (e.g. page 10 lines 10-12) this is somewhat confusing. As a result there is the potential for a mixed message. I don't think the data allow a specific test of the safe harbour hypothesis, which in any case is somewhat over simplistic. Instead I would be inclined to broaden it out to recognise that ecological factors more generally are associated with the initial evolution of parental care but subsequent acceleration of the rate of phenotypic evolution is facilitated by appropriate behavioural precursors and parent-offspring trait coadaptation (see e.g. Royle et al 2016 Current Opinion in Behavioral Sciences for a review). This would, in my opinion, better frame the work and largely avoid getting in a tangle about harshness or otherwise of environments promoting parental care.

In addition, to broaden out the appeal of the paper still further it would also be good to add a little bit of text in the discussion that considers the broader potential implications of the work for other taxa such as birds where parental care is more ubiquitous but where there is less variation in forms of care than amphibians, or in fishes. Is the effect of parental care on the evolution of clutch size relative to egg size likely to be less marked in birds given the smaller variation in clutch size and the greater complexity of parental care for example?

Specific minor comments

Introduction

P3

The first paragraph of the introduction is big on bigging up the work up but leaves many basic Qs unanswered which is very frustrating rather than enticing for this reader in any case.

9-10: for example?

11-13: an example or two would be useful here too.

16-20: there are two proposed drivers, so what is the difference in the prediction in terms of egg and clutch size if any? This sentence is overly vague and unspecified. Having read through the MS I know what you mean but it needs to be clearer up front.

21-22: it is still unclear what exactly it is about the new Bayesian approach that allows causation to be identified. Makes for a frustrating read at this stage.

23: what, even under biparental care? Needs clarification.

P4

17-21: so…? Isn't this a given if we accept that environmental conditions drive egg size evolution and there is a trade-off between egg size and clutch size? 

22: are they 'opposite' conclusions or just different/variable conclusions?

P5

5-7: difficult to understand sentence. Rewrite for clarity?

P6

15-16: independently? How does this demonstrate causation?

P7

10-16: very well explained.

18-20: ok, but that doesn't allow you to differentiate between PC and habitat as drivers does it? Would help to have a little further explanatory info here. 

Results

P8

14-19: could these be illustrated in a figure? It would be really useful if done well.

21: wouldn't just saying "body size" be more accurate here than "allometry", given that was the term used in the models? 

P9

11: by "terrestrial offspring" do you mean both eggs and larvae? It is a bit confusing to introduce a new term here.

Discussion

P13

7-8: this is a really nice point but could you also broaden it out to make some predictions about other taxa such as birds or fish too, for example?

Methods

P15

12-14: how many fit this category? Info would be useful here, or refer to table.

P18

15: wow. That's a lot of iterations. Why so many?

25: is direct development a binary variable then?

P19

1: terrestrial eggs and larvae also binary variables?

Figure 2 - for a) and d) it would be useful to have "egg size" and "clutch size" respectively at the top of the posterior distributions to make it easier to see what is what.

Figure 3 - a) there is crazy amounts of stuff going on here. It is very difficult to make much sense of it without assistance (e.g. highlighting key areas or changing the scale for some of the variables) or simplification. The figure looks good at a distance but it is too complicated in my opinion. b) this is very nice and simple and well explained on the other hand….

Table S2 - some of these variables have very small sample sizes e.g. tadpole traits, so it would be worth noting a word of caution in the results at an appropriate juncture to help interpret the effect sizes, which are quite large even for non-significant results. 

Table S5 - it would be good to have a brief summary in the main text at an appropriate juncture of some of the key clades/species that have very high rates of evolution that are given here. It would help to ground the study and make it more interesting and accessible. 

Reviewer #3:

In this manuscript the authors analyze the relationship between offspring size and number among 800 sampled amphibian species. They find (as do previous studies) that species that lay eggs on land (including those with direct development) have larger eggs and smaller clutch sizes. They also find that different forms of parental care (e.g. male vs. female, care of egg vs. larvae) can have different effects on egg and clutch sizes, including both increases and decreases. 

The authors have assembled a relatively large dataset for this study, building on their previous studies and those of other authors. They also analyze their data using a relatively recent (2016) approach that identifies shifts in rates of phenotypic evolution and identifies the correlates of those shifts.

I think that this is a very good manuscript, and I have almost no specific criticisms to make (which is very unusual for me). On the negative side, I do not think that it is of sufficiently broad interest or novelty for a top-tier journal like PLoS Biology. Basically, it is clearly an improvement over previous studies on this relatively narrow topic, and adds nuance to their conclusions. That is certainly enough to justify its existence, but not necessarily its publication in a top-tier journal. This seems more like a paper for Am. Nat., Evolution, or Proceedings B. However, I will admit that this is a relatively subjective decision.

COMMENTS FROM THE ACADEMIC EDITOR [lightly edited]:

Ref 1 is the most critical reviewer. I agree with a lot of what they say. However, I think what they say is a general criticism of ALL comparative analyses. They use the maximum data available to report correlations between traits. Thus, in some ways, they are purely descriptive exercises. They can be treated as hypothesis testing at one level if one choses to state the direction of the correlation in advance, but a post-hoc attempt to now only look at a subset of predictors would be disingenuous. The authors already know where the strongest relationships lie. That said, the ref is right that the MS can be improved by more clearly stating which predictors are highly correlated and, perhaps, picking one of several that are highly correlated as the 'representative' of a suite of correlated predictors. They are also right that the authors need to better explain why this MS is an advance over Lim et al. 2014.

Ref 1 is focused on the general question which they feel has been answered in marine inverts. Perhaps this is true, but it is common for people working on different taxa to want to test relationships in their group. I think the paper will be of interest to herpetologists and that is actually sufficient to justify publication. (After all lots of bird studies are published simply because someone again shows something in birds that we already know occurs in insects, likewise for marine vs terrestrial). Crucially, I do not think the marine studies cited have the equivalent 'add on' of testing how parental care moderates the offspring size-number tradeoff.

In sum, I would encourage a substantive revision that takes into account the reviewers' criticisms; which is essentially the minor points of Ref 2 and the 'reduce the number of highly correlated predictors' concern of Ref 1. Ref 3 did not ask for any changes.

---

## [Editor Report · Decision Letter 2]

17 Nov 2021

Dear Dr Capellini,

Thank you for submitting your revised Research Article entitled "Terrestrial reproduction and parental care drive rapid adaptation in the offspring size-number tradeoff across amphibians" for publication in PLOS Biology. The Academic Editor has now kindly checked your revisions and responses to the reviewers' previous comments, thereby saving us a further round of review.

Based on the Academic Editor's assessment, we will probably accept this manuscript for publication, provided you satisfactorily address the following data and other policy-related requests.

a) Please make your title more accessible to those outside the field. We suggest: "Terrestrial reproduction and parental care drive rapid evolution in the tradeoff between offspring size and number across amphibians"

b) Please address my Data Policy requests below; specifically, while we recognise that you’ll be depositing the raw data in Dryad, we need you to supply numerical values underlying Figs 2ABCDEF, 3A. Please cite the location of the data clearly in each relevant Fig legend.

We expect to receive your revised manuscript within two weeks. 

*Published Peer Review History*

*Early Version*

Sincerely,

Roli Roberts

Senior Editor,

rroberts@plos.org,

PLOS Biology

DATA POLICY:

We note that your raw data will be deposited in Dryad. However, we also need the numerical values displayed in the figures of your paper be made available in one of the following forms:

Regardless of the method selected, please ensure that you provide the individual numerical values that underlie the summary data displayed in the following figure panels as they are essential for readers to assess your analysis and to reproduce it: Figs 2ABCDEF, 3A. NOTE: the numerical data provided should include all replicates AND the way in which the plotted mean and errors were derived (it should not present only the mean/average values).

DATA NOT SHOWN?

---

## [Editor Report · Decision Letter 3]

26 Nov 2021

Dear Isabella,

On behalf of my colleagues and the Academic Editor, Michael Jennions, I'm pleased to say that we can in principle accept your Research Article "Terrestrial reproduction and parental care drive rapid evolution in the tradeoff between offspring size and number across amphibians" for publication in PLOS Biology, provided you address any remaining formatting and reporting issues. These will be detailed in an email that will follow this letter and that you will usually receive within 2-3 business days, during which time no action is required from you. Please note that we will not be able to formally accept your manuscript and schedule it for publication until you have any requested changes.

PRESS: We frequently collaborate with press offices. If your institution or institutions have a press office, please notify them about your upcoming paper at this point, to enable them to help maximise its impact. If the press office is planning to promote your findings, we would be grateful if they could coordinate with biologypress@plos.org. If you have not yet opted out of the early version process, we ask that you notify us immediately of any press plans so that we may do so on your behalf.

Sincerely,

Roli 

Roland G Roberts, PhD 

Senior Editor 

PLOS Biology

rroberts@plos.org